# A unique Co@CoO catalyst for hydrogenolysis of biomass-derived 5-hydroxymethylfurfural to 2,5-dimethylfuran

Shuang Xiang[1,6], Lin Dong[1,6], Zhi-Qiang Wang[1,6], Xue Han[2], Luke L. Daemen[3], Jiong Li[4], Yongqiang Cheng[3], Yong Guo[1], Xiaohui Liu[1], Yongfeng Hu[5], Anibal J. Ramirez-Cuesta[3], Sihai Yang[2✉], Xue-Qing Gong[1✉] & Yanqin Wang[1✉]

The development of precious-metal-free catalysts to promote the sustainable production of fuels and chemicals from biomass remains an important and challenging target. Here, we report the efficient hydrogenolysis of biomass-derived 5-hydroxymethylfurfural to 2,5-dimethylfuran over a unique core-shell structured catalyst, Co@CoO that affords the highest productivity among all catalysts, including noble-metal-based catalysts, reported to date. Surprisingly, we find that the catalytically active sites reside on the shell of CoO with oxygen vacancies rather than the metallic Co. The combination of various spectroscopic experiments and computational modelling reveals that the CoO shell incorporating oxygen vacancies not only drives the heterolytic cleavage, but also the homolytic cleavage of $H_2$ to yield more active $H^{\delta-}$ species, resulting in the exceptional catalytic activity. Co@CoO also exhibits excellent activity toward the direct hydrodeoxygenation of lignin model compounds. This study unlocks, for the first time, the potential of simple metal-oxide-based catalysts for the hydrodeoxygenation of renewable biomass to chemical feedstocks.

[1] Key Laboratory for Advanced Materials and Joint International Research Laboratory of Precision Chemistry and Molecular Engineering, Feringa Nobel Prize Scientist Joint Research Center, Research Institute of Industrial Catalysis, School of Chemistry and Molecular Engineering, East China University of Science and Technology, Shanghai 200237, China. [2] Department of Chemistry, University of Manchester, Manchester M13 9PL, UK. [3] Neutron Scattering Division, Neutron Sciences Directorate, Oak Ridge National Laboratory, Oak Ridge, TN 37831, USA. [4] Shanghai Synchrotron Radiation Facility, Shanghai Advanced Research Institute, Chinese Academy of Sciences, Shanghai 201210, China. [5] Sinopec Shanghai Research Institute of Petrochemical Technology, Shanghai 201208, China. [6] These authors contributed equally: Shuang Xiang, Lin Dong, Zhi-Qiang Wang. ✉email: Sihai.Yang@manchester.ac.uk; xgong@ecust.edu.cn; wangyaqin@ecust.edu.cn

**B**iomass is the only renewable resource of organic carbons in nature and their conversion to value-added chemicals and liquid fuels is of vital importance in achieving global carbon neutralisation[1,2]. Cellulose-derived 5-hydroxymethylfurfural (HMF) is widely recognised as a platform chemical for the synthesis of sustainable liquid fuels and chemicals[3,4]. Particularly, the selective hydrogenolysis of HMF to 2,5-dimethylfuran (DMF) as biofuels and feedstocks of renewable p-xylene has attracted much interest[5,6]. A great deal of effort has been devoted to developing supported metal catalysts for this reaction, and state-of-the-art catalysts are based upon Ru, Pd, Pt, Ni and Cu materials[7–11]. We have designed cobalt oxide-supported ruthenium (Ru/Co$_3$O$_4$) and cobalt/nickel [(Co)Ni/Co$_3$O$_4$] catalysts that show DMF yields of 93% and 70–76%, respectively, at 130 °C for 24 h[7,8]. Schüth et al. developed a hollow platinum-cobalt bimetallic nanoparticle (PtCo@HCS) catalyst, which achieved a high yield of DMF (98%) at 180 °C for 2 h[10]. Esteves et al. investigated various supported copper catalysts and identified a high yield of DMF (93%) over Cu/Fe$_2$O$_3$-Al$_2$O$_3$ at 150 °C for 10 h[11]. To date, metal and harsh reaction condition (i.e., high temperature and/or long reaction time) are almost indispensable to achieve the high yield of DMF. It is widely accepted that the homolytic dissociation of H$_2$ occurs on these metal catalysts, generating free radicals (H·) to drive the subsequent hydrogenolysis[12]. Recently, it is reported that H$^{\delta-}$ species obtained via heterolytic dissociation of H$_2$ showed enhanced catalytic performance[13–17]. Thus, the development of new catalysts that can generate H$^{\delta-}$ species hold great promise to promote the hydrogenolysis of HMF under mild reaction conditions.

Although single-atom catalysts can catalyse the heterolytic cleavage of H$_2$[13–15], there is complicity associated with their preparation and thermodynamic stability. Meanwhile, metal oxides with a high concentration of surface defects are reported as emerging catalysts with high activity for the heterolytic cleavage of H$_2$[16–22]. For example, ceria with oxygen vacancies (O$_v$) can produce H$^{\delta-}$ species via the heterolytic pathway and showed excellent activity in hydrogenation reactions[16,17], where the oxygen vacancies played an important role in the formation and stabilisation of hydride species (Ce$_{Ov}^{4+}$-H$^-$)[18]. However, metal oxides that can promote hydrogenolysis via integrated homolytic and heterolytic cleavage of H$_2$ have not been reported to date.

Here, we report an unusual core-shell structured catalyst, Co@CoO, which can promote the integrated homolytic and heterolytic cleavage of H$_2$, affording an exceptional performance for the hydrogenolysis of HMF to DMF under mild conditions. Importantly, a superior productivity of DMF (17.58 mmol·g$^{-1}$ h$^{-1}$) was achieved over Co@CoO at 130 °C. Moreover, Co@CoO demonstrated an excellent catalytic stability of over 100 hours in a continuous flow reaction at a space velocity as high as 26.6 h$^{-1}$. Transmission electron microscopy (TEM), X-ray photoelectron spectroscopy (XPS), inelastic neutron scattering (INS) and density functional theory (DFT) calculations confirm that the superior catalytic performance is attributed to the CoO shell decorated with oxygen vacancies. It not only catalyses the homolytic/heterolytic cleavage of H$_2$ to generate H$^{\delta-}$ species, but also promotes the adsorption and activation of HMF. Co@CoO also exhibits an excellent performance for the hydrogenolysis of the lignin β-O-4 model compound. This study will inspire the design of new efficient catalysts based upon precious-metal-free metal oxides to promote the synthesis of renewable biofuels and chemicals.

## Results

**Hydrogenolysis of HMF to DMF.** The hydrogenolysis of HMF was firstly conducted in a batch reactor at 130 °C with 1 MPa H$_2$ for 2 h over a series of Co$_3$O$_4$-temp. (temp. = temperature for reduction in °C) catalysts, which are prepared by a simple precipitation method followed by reduction in 10% H$_2$ for 2 h (see Methods). It is reported that there are two possible pathways for the hydrogenolysis of HMF to DMF over supported metal catalysts (Fig. 1). Path 1 proceeds through the hydrogenation of C=O to C–OH group to give 2,5-furandimethanol (BHMF), which is

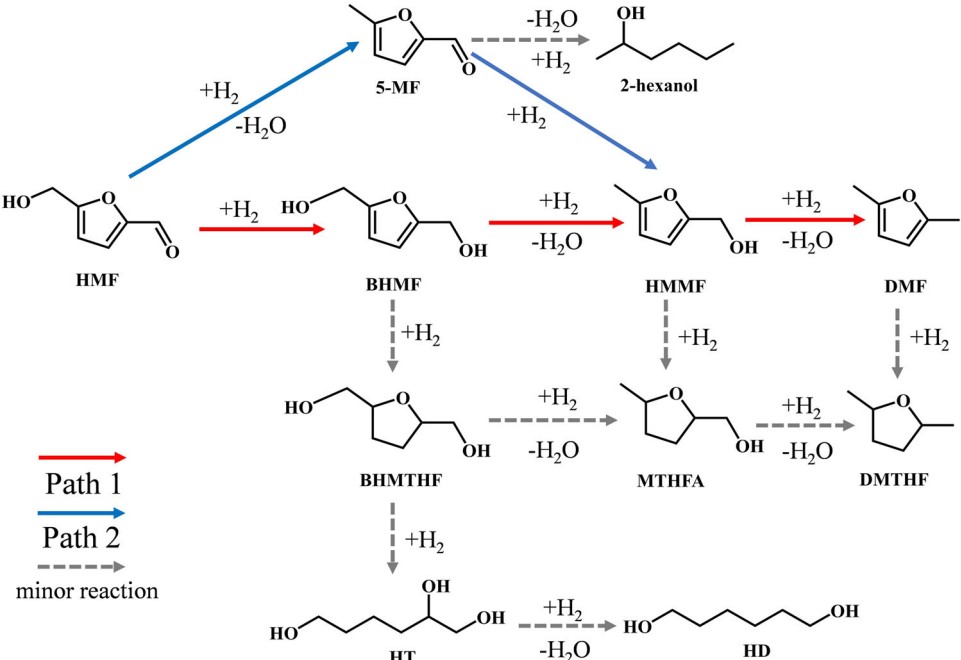

**Fig. 1 The reaction pathways and network for the hydrogenation/hydrogenolysis of HMF to DMF.** Path 1. HMF is first hydrogenated to BHMF, than converted to DMF step-by-step through hydrogenolysis, with BHMF and HMMF as the intermediates. Path 2: HMF is converted to 5-MF through hydrogenolysis, followed by hydrogenation and further hydrogenolysis to DMF, with 5-MF and HMMF as the intermediates. Minor reaction: HMF is first hydrogenated to BHMF, followed by the saturation of furan ring in each step, even with the opening of tetrahydrofuran ring.

**Table 1 Summary of results of conversion of HMF over different Co-based catalysts[a].**

| Catalyst | Conv. (%) | Yield (%) | | | | DMF productivity/mmol·g$^{-1}$h$^{-1}$ |
|---|---|---|---|---|---|---|
| | | BHMF | HMMF | DMF | Others[b] | |
| Co$_3$O$_4$ | 1.9 | 0.0 | 0.0 | 0.0 | 0.0 | 0.0 |
| Co$_3$O$_4$-200 | 2.4 | 0.0 | 0.0 | 0.0 | 0.0 | 0.0 |
| Co$_3$O$_4$-250 | >99 | 0.0 | 4.3 | 89.2 | 0.0 | 17.6 |
| Co$_3$O$_4$-300 | >99 | 0.0 | 54 | 39.0 | 1.2 | 7.7 |
| Co$_3$O$_4$-400 | 91 | 65.0 | 16 | 2.9 | 1.3 | 0.5 |
| 0.5%Ru/Co$_3$O$_4$ | >99 | 1.2 | 11.8 | 69.8 | 0.6 | 13.7 |
| 5.0%Ru/Co$_3$O$_4$ | 95.0 | 21.3 | 25.8 | 22.4 | 4.9 | 4.4 |
| Commercial CoO | 1.2 | 0.0 | 0.0 | 0.0 | 0.0 | 0.0 |
| Commercial CoO-reduced | 60.0 | 45.0 | 4.3 | 1.1 | 2.2 | 8.9 |

[a]Reaction conditions: HMF (150 mg), catalyst (30 mg), THF (5 mL), H$_2$ pressure (1 MPa), temperature (130 °C), 2 h.
[b]Others mainly include 5-methyl furfural (5-MF) and other unidentified products.

followed by the hydrogenolysis of C–OH groups to 5-methyl-2-furanmethanol (HMMF) and finally to DMF[5,6]. Path 2 undergoes the hydrogenolysis of HMF to 5-methyl furfural (5-MF), followed by the hydrogenation of 5-MF and hydrogenolysis of HMMF to DMF[23,24]. Other reactions include the hydrogenation and/or opening of furan-ring (marked as minor reaction in Fig. 1) as well as the condensation. The reaction pathway over Co$_3$O$_4$-250 followed the reported Path 1 and HMMF was the main intermediate (Supplementary Fig. 1).

Table 1 shows that among all the investigated catalysts, Co$_3$O$_4$-250 shows the best catalytic performance and the yield of DMF reaches 89.2% on full conversion of HMF. Unreduced Co$_3$O$_4$ and Co$_3$O$_4$-200 catalysts give no activity. Co$_3$O$_4$-300 and Co$_3$O$_4$-400 show poor activity with HMMF and BHMF being the main product, respectively. The GC-MS plots are shown in Supplementary Figs. 2–4. These differences in activity clearly indicates that the extent of reduction of Co$_3$O$_4$ plays a key role in determining its structure and hence the activity of the catalysts. The influence of solvents was investigated and THF shows the best performance (Supplementary Table 1 and Supplementary Fig. 5). Interestingly, over these Co$_3$O$_4$-temp. catalysts, the carbon balance is all above 90%. This is probably due to the neutrality of Co@CoO, which prevents the undesired condensation of intermediates/products that is catalysed by acidic or basic sites.

A small amount of HMMF (4.3%) was detected over Co$_3$O$_4$-250 in the first 2 h of the reaction, and this can be further converted into DMF with a total yield of 92.4% in 3 h. At 100 °C, a DMF yield of 54.9% was obtained over Co$_3$O$_4$-250 in 6 h (Supplementary Table 2). At 150 and 180 °C, the reaction was conducted with twice amount of the substrate of HMF due to the accelerated reaction kinetics and the yield of DMF was 73 and 53%, respectively (Supplementary Table 2). The productivities are calculated (Fig. 2a) and importantly, Co$_3$O$_4$-250 shows remarkably high productivities across the temperature range of 100–250 °C, outperforming all state-of-the-art metal-based catalysts[7,8,10,11,25–42], including noble-metal-based catalyst. To further confirm the super activity of Co$_3$O$_4$-250, 0.5 and 5%Ru-loaded Co$_3$O$_4$ catalysts prepared by impregnation and reduction were also used in this reaction and the results were added in Table 1. The productivity is decreased from 17.6 over Co$_3$O$_4$-250 to 13.7 and 4.4 mmol DMF·g$^{-1}$h$^{-1}$ over 0.5 and 5%Ru-loaded Co$_3$O$_4$ catalysts, respectively. These results clearly show that the

addition of Ru hinder the reaction and the performance of 5%Ru is even worse than that of 0.5%Ru, which hint that the active site is unique over Co$_3$O$_4$-250 catalyst.

The excellent stability of Co$_3$O$_4$-250 for the conversion of HMF to DMF has been demonstrated by a continuous flow reaction over 100 h (Fig. 2b). To further examine the catalytic stability, the weight hourly space velocity (WHSV) is increased to 26.6 h$^{-1}$, much higher than that (3.3 h$^{-1}$) over 2%Ni-20%Co/C catalyst[35] and other reports[43,44]. Significantly, Co$_3$O$_4$-250 shows excellent activity and stability with little decrease in DMF yield (>75%). In addition, TEM images suggest an absence of notable structural change of Co$_3$O$_4$-250 post the 100 h time-on-stream test (Supplementary Fig. 6).

**Catalyst characterisation.** To investigate the effect of reduction temperature to the structure of the catalysts, high-resolution TEM images were taken with all Co-based catalysts, and significant differences in composition and crystal structure were observed (Fig. 3a, b). The unreduced Co$_3$O$_4$ catalyst shows the interplanar crystal spacing of 0.285 nm, corresponding to the (220) plane of the spinel-structured Co$_3$O$_4$. While Co$_3$O$_4$-200 and Co$_3$O$_4$-250 both show core-shell structures, the former is CoO@Co$_3$O$_4$ and the latter is Co@CoO. Importantly, Co$_3$O$_4$-250 has a tight CoO shell and this core-shell structure was observed uniformly over the Co$_3$O$_4$-250 sample (Fig. 3b and Supplementary Fig. 7). While over the Co$_3$O$_4$-300 catalyst, the CoO shell becomes thinner and generates a minor amount of metallic Co on the surface. These results indicate that the reduction of Co$_3$O$_4$ may take place from the core, and the phase composition and morphology are evolved from spherical Co$_3$O$_4$ to core-shelled CoO@Co$_3$O$_4$, then to core-shelled Co@CoO with different thickness of the shell.

XRD analysis was also conducted to characterise the phases of these catalysts (Supplementary Fig. 8). For unreduced Co$_3$O$_4$, only spinel-phase Co$_3$O$_4$ (PDF#42-1003) was observed. The main phase of Co$_3$O$_4$-200 is spinel-phase Co$_3$O$_4$ with a small amount of cubic-phase CoO (PDF#48-1719). The XRD patterns of Co$_3$O$_4$-250 and Co$_3$O$_4$-300 show a mixture of cubic phases of Co and CoO. Co$_3$O$_4$-400 shows a primarily metallic Co structure with little catalytic activity.

The Co K-edge X-ray absorption spectra (XAS) of Co$_3$O$_4$-200, Co$_3$O$_4$-250 and Co$_3$O$_4$-300 have been measured and the spectra in R space are shown in Supplementary Fig. 9 with Co foil, CoO and Co$_3$O$_4$ as references. The local environment of Co in Co$_3$O$_4$-

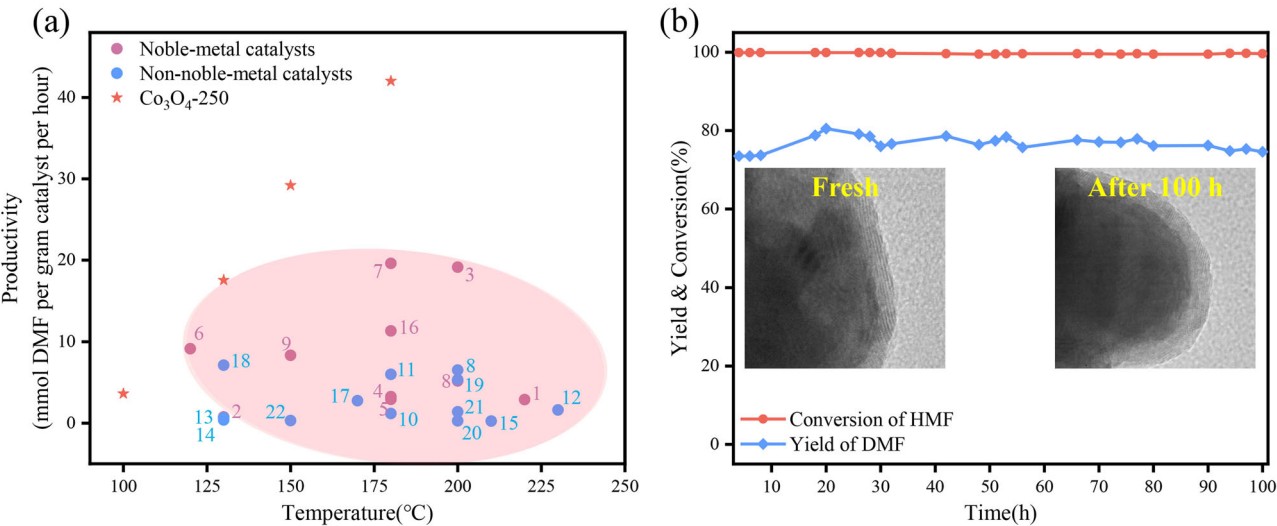

**Fig. 2 Catalytic performance of Co₃O₄-250 in the hydrogenolysis of HMF to DMF. a** Comparison of the catalytic performances of the state-of-the-art catalysts for the hydrogenolysis of HMF to DMF. A summary of the productivities of representative noble-metal catalysts (purple circles), non-noble-metal catalysts (blue circles), Co₃O₄-250. Full data is shown in Table S1. (1) Ru-doped hydrotalcite[25]; (2) Ru/Co₃O₄[7]; (3) RuCo/CoOₓ[26]; (4) Ru/CoFe-LDO[27]; (5) Pt₁/Co[28]; (6) Pt/rGO[29]; (7) PtCo@HCS[10]; (8) Pd-OMD1/Ni-OMD1[30]; (9) Pd/C/Zn[31]; (10) Raney Co[32]; (11) Ni/C[33]; (12) Ni/LaFeO₃[34]; (13) Ni-Co oxides[8]; (14) Ni-Co/C[35]; (15) 2%Ni-20%Co/C[36]; (16) Ag-Co@C[37]; (17) Co-CoOₓ[38]; (18) Co/Mix-ZrO₂[39]; (19) CuZn[40]; (20) Cu-Ni/Al₂O₃[41]; (21) CuNi/TiO₂[42]; (22) Cu/Fe₂O₃-Al₂O₃[11]. **b** Catalytic performance and stability of the hydrogenolysis of HMF to DMF. Reaction conditions: 130 °C, 1 MPa of H₂, 26.6 h⁻¹ WHSV and 30 mL min⁻¹ H₂ gas flow rate.

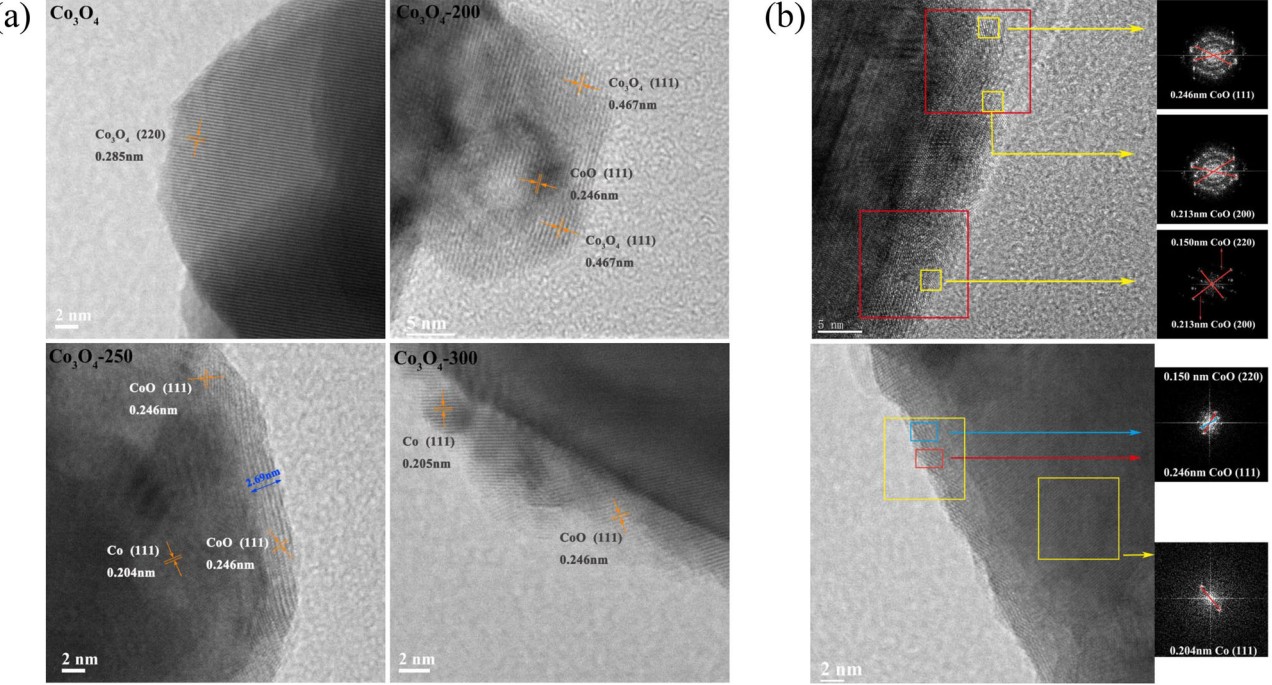

**Fig. 3 HRTEM images of various catalysts. a** HRTEM images of selected catalysts (unreduced Co₃O₄, Co₃O₄-200, Co₃O₄-250 and Co₃O₄-300); **b** representative HRTEM images of Co₃O₄-250. The HRTEM images of Co₃O₄-250 clearly show the closed shell of CoO, with the thickness of 2–3 nm and CoO(111), CoO(200) dominated.

250 and Co₃O₄-300 is similar to that of Co foil[45,46] with a dominant Co–Co feature at 2.18 Å. Compared with the Co foil, the amplitude of this feature is much lower for the Co₃O₄-250 and Co₃O₄-300, indicating the metallic Co in these two catalysts are not as well ordered as metallic Co. Meanwhile, there is no metallic Co–Co feature at 2.18 Å for Co₃O₄-200, which is similar to that of CoO with peaks around 1.45 and 2.60 Å, corresponding to the Co–O and Co–Co distance, respectively. In addition, a weak signal at 2.70 Å is observed over Co₃O₄-250, similar to the Co–O bond in the CoO standard, further confirming that the sample is not reduced completely. These results suggest that the Co species on Co₃O₄-200 is CoO, and the metallic Co and CoO species co-exist in Co₃O₄-250, which are in agreement with the HRTEM results.

To further explore the electronic properties of surface Co species, XPS spectra of all samples were recorded (Fig. 4a). The

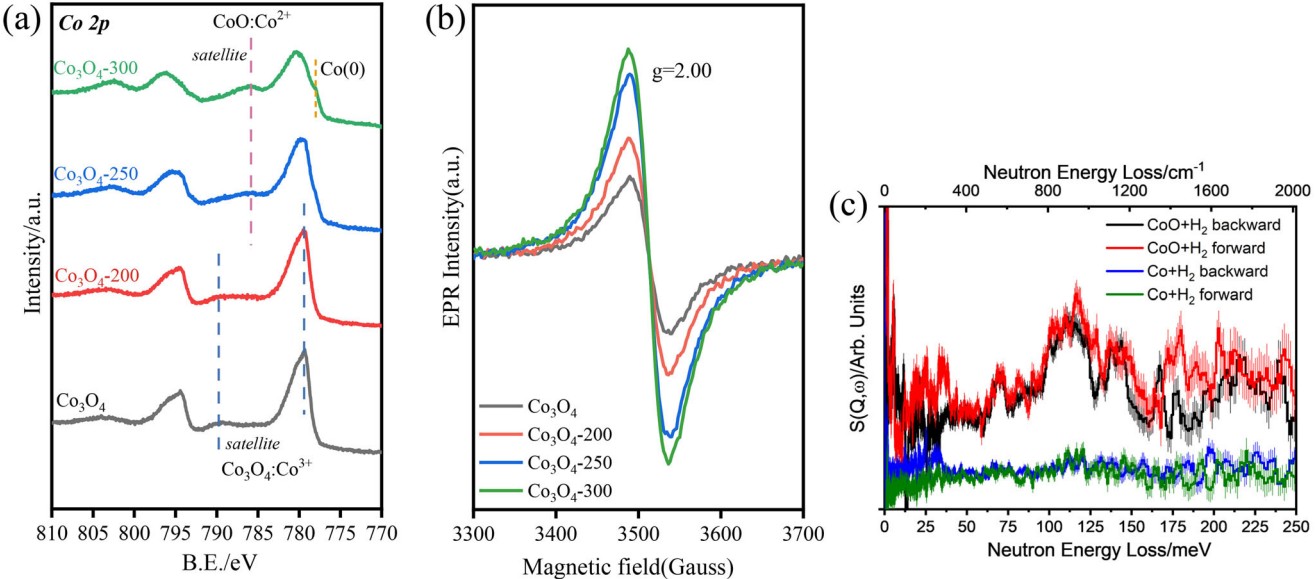

**Fig. 4 Characterizations of catalysts with various techniques. a** XPS spectra of the Co 2p orbital and **b** X-band EPR spectra of unreduced $Co_3O_4$, $Co_3O_4$-200, $Co_3O_4$-250 and $Co_3O_4$-300 at 77 K. **c** Comparison of the difference INS spectra of the $Co_3O_4$-250 and Co catalysts upon hydrogenation showing unique signals of hydrogenous species on $Co_3O_4$-250. The difference spectra are obtained by subtracting the INS spectrum of fresh catalyst from that of the hydrogenated catalyst.

binding energy, shape and intensity of the satellite peaks are used to identify the cobalt species[47,48], because the satellite peaks associated with the Co $2p_{3/2}$ peaks of $Co_3O_4$ and CoO are distinct[49]. $Co_3O_4$ and $Co_3O_4$-200 both exhibit peaks at 779.9 eV and 796.7 eV, which are attributed to $Co^{3+}$ of $Co_3O_4$. While $Co_3O_4$-250 shows a strong satellite peak of Co $2p_{3/2}$ at 786.0 eV, indicating the presence of CoO on the surface. An additional peak appeared at 778.0 eV for $Co_3O_4$-300, which belongs to metallic Co, confirming the coexistence of both metallic Co and CoO on the surface. Thus, the XPS study confirms the presence of a shell of CoO on $Co_3O_4$-250.

It is reported that the concentration and location of oxygen vacancy in $CeO_2$ play an important role in its activity for hydrogenation reactions[16–18]. Commercial CoO and its pre-reduced analogue (250 °C, 10 min) were used to further clarify the importance of surface defects, and the results are summarized in Table 1. Negligible conversion was observed with commercial CoO, but with the reduced commercial CoO, the yield of BHMF and DMF is 45.0% and 1.1%, respectively. To study the oxygen vacancy of these Co-based catalysts, X-band continuous wave electron paramagnetic resonance (EPR) spectra were collected at 77 K. Compared with $Co_3O_4$-200, the signal at $g = 2.00$ appeared and is gradually enhanced upon increase of the reduction temperature from 200 to 300 °C, demonstrating the increased concentration of oxygen vacancies (Fig. 4b).

In situ INS was conducted to examine the formation of cobalt hydride upon the activation of $H_2$ over $Co_3O_4$-250. Comparison of the difference of INS spectra before and after the dissociation of $H_2$ over $Co_3O_4$-250 (that is, signals for intermediates that may form under the conditions of the hydrogenation reaction) showed a number of marked changes (Fig. 4c). The main feature (I) centred at 110 meV evidently indicates the formation of Co–H species[50]. A broad underlining intensity across 80–160 meV is also observed, and this can be assigned to the formation of Co–O–H moieties; a similar feature at 75–150 meV has been observed previously upon the formation of Fe–O–H (75–150 meV) species during the heterolytic dissociation of $H_2$ over a $CuCrFeO_x$ catalyst[51]. The sharp features above 160 meV are unlikely caused by any H-containing species formed under

reaction conditions because of the inconsistent intensities detected for this region by the forward and backward detectors (Fig. 4c). These results hint heterolytic splitting of $H_2$ has occurred. To gain further insights, the in situ INS experiment was also carried out with the metallic Co catalyst ($Co_3O_4$-400), and upon reacting with $H_2$ under same conditions, no distinct features were observed for $Co_3O_4$-400, indicating that the presence of surface Co–O and O vacancy is crucial for the formation Co–H intermediates. The raw spectra were presented in Supplementary Figs. 10 and 11. Therefore, the unique Co@CoO core-shell structure of $Co_3O_4$-250 with rich oxygen vacancy directly promotes the formation of $H^{\delta-}$ species.

**Density functional theory studies.** To elucidate the excellent activity and importance of oxygen vacancy on the CoO shell of $Co_3O_4$-250 for the hydrogenolysis of HMF, theoretical investigations using electronic density functional theory (DFT) method were carried out. All calculations were performed with Vienna Ab-initio Simulation Package (VASP) (see Methods). We first built the $p(2 \times 3)$ surface slabs with six atom layers for the stoichiometric CoO(100) and CoO(100)-Ov surfaces. The calculated oxygen vacancy formation energy of the CoO(100) surface is 5.77 eV (Supplementary Fig. 12). Then, the energy profiles of the adsorption and dissociation of $H_2$ on the CoO(100) and CoO(100)-Ov surfaces were calculated (Fig. 5a, Supplementary Fig. 13 and Supplementary Table 3). Firstly, the adsorption energies of $H_2$ at CoO (100) and CoO(100)-Ov were calculated to be 0.25 and 1.63 eV, respectively, indicating that CoO(100)-Ov possesses a stronger binding ability. Subsequently, the dissociation of $H_2$ in a heterolytic way was calculated. This process is endothermic by 0.54 eV and gives a barrier of 0.60 eV over CoO(100), while on the CoO(100)-$O_V$ surface it is less endothermic (0.36 eV) and with a lower barrier of 0.56 eV. Therefore, the CoO(100)-$O_V$ appears to be more beneficial for $H_2$ dissociation in the heterolytic way. For more comprehensive understanding, we also calculated the homolytic dissociation of $H_2$ on the CoO(100)-$O_V$ surface (Supplementary Fig. 13). Surprisingly, the $H_2$ on the CoO(100)-$O_V$ surface was split into two

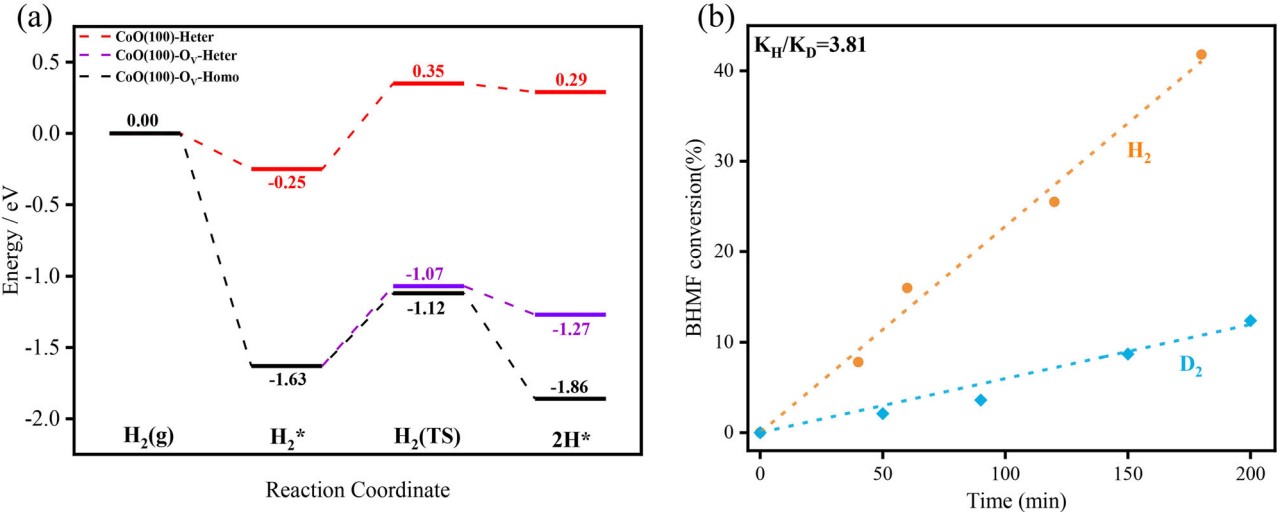

**Fig. 5 Theoretical calculation and kinetic studies. a** Calculated energy profiles of adsorption and dissociation of $H_2$ on the CoO(100) and CoO(100)-$O_V$ surfaces. $H_2$(g): gas-phase $H_2$; $H_2$*: adsorbed $H_2$ on surface; $H_2$(TS): the adsorbed $H_2$ molecule on surface dissociates to two adsorbed H on surface; 2H*: the co-adsorption of two H on surface; **b** primary kinetic isotope effect observed for the HDO of 2,5-furandimethanol (BHMF). Reaction condition: BHMF, 200 mg; catalyst ($Co_3O_4$-250), 20 mg; THF, 5 mL; temperature, 130 °C; $H_2/D_2$, 1 MPa.

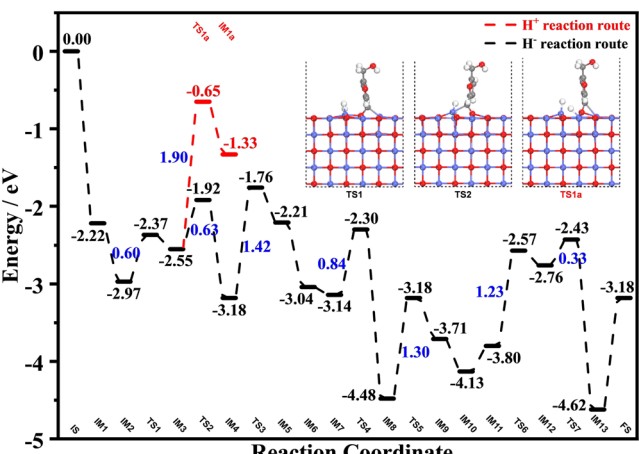

**Fig. 6 Calculated energy profiles of HMF hydrogenolysis reaction on the CoO(100)-$O_V$ surface.** Red dotted line refers to the $H^+$ route, black dotted line is the $H^-$ route. The corresponding energy barriers ($E_a$, blue) for key steps are also included (in eV). The details in each separate step is provided in Supplementary Fig. 19.

$H^{\delta-}$ with the help of the $O_V$, as confirmed by the Bader charge analysis (Supplementary Fig. 13), and such homolytic dissociation of $H_2$ is calculated to be exothermic (0.23 eV) and with an energy barrier of 0.51 eV. This indicates that the homolytic dissociation is easier than the heterolytic one and the formation of two $H^{\delta-}$ may make the CoO(100)-$O_V$ surface more active than CoO(100).

To illustrate the catalytic activities of different facets, the energy profiles of the adsorption and dissociation of $H_2$ on the CoO(111) and CoO(111)-$O_V$ surfaces have been calculated (Supplementary Figs. 14 and 15). The homolytic dissociation of $H_2$ is exothermic by 3.32 eV and gives a barrier of 0.91 eV over CoO(111). On the CoO(111)-$O_V$ surface, it is also exothermic (3.56 eV), but the barrier is significantly reduced to 0.32 eV. This result indicates that CoO(111)-$O_V$ is more beneficial than CoO(111) for $H_2$ dissociation, in agreement with what was determined on the CoO(100) and CoO(100)-Ov surfaces. Notably, the CoO(111) surface is a polar surface with O-terminated structure, which tends to form stable OH

species. Nonetheless, $H_2$ on the CoO(111)-$O_V$ surface can still be split into one $H^{\delta+}$ and one $H^{\delta-}$ with the help of the $O_V$, as confirmed by the Bader charge analysis (Supplementary Fig. 15). Such heterolytic dissociation of $H_2$ was calculated to be exothermic (1.59 eV) with an energy barrier of 0.41 eV. In comparison, the CoO(100)-$O_V$ surface is more favourable to produce a large number of active $H^{\delta-}$ species due to the strong adsorption of $H_2$ (1.63 eV), and the $H^{\delta-}$ species can be also produced in a homolytic way. Thus, all these results show that the hydride can be produced by $H_2$ dissociation on both the reduced CoO(111) and Co(100) surfaces.

The adsorption structures and energetics of HMF on the CoO(100) and CoO(100)-Ov surfaces have been calculated and give the following findings (Supplementary Fig. 16). (i) HMF can be parallelly and vertically adsorbed on the two surfaces via C=O or C–OH groups. The two possible bridging adsorption structures were determined and the calculated adsorption energies at CoO(100) are 0.71 and 0.47 eV (Supplementary Fig. 16a, b), respectively. While on the CoO(100)-$O_V$ surface, the adsorption energies are 1.03 and 1.92 eV (Supplementary Fig. 16l, h), respectively. (ii) The calculated adsorption energies at CoO(100)-$O_V$ are higher than those at CoO(100) surface, indicating that the CoO(100)-$O_V$ possesses a stronger binding ability. (iii) The calculated highest adsorption energy of HMF on the CoO(100) surface is only 0.71 eV, in agreement with the poor activity of the commercial CoO catalysts, and these results are consistent with the above mentioned study[52]. (iv) The adsorption energy determined for the interaction of HMF with the surface through the C=O group is the strongest (2.22 eV) among all those determined on the CoO(100)-Ov surface, which may explain the improved performance of the first hydrogenation step of HMF to BHMF, followed by hydrogenolysis to HMMF and DMF over the $Co_3O_4$-250 catalyst. In addition, we also calculated the Bader charges of the atoms in the adsorbed HMF on the CoO(100) and CoO(100)-$O_V$ surfaces. The results showed that the stronger the HMF adsorption is, the more electrons the HMF can obtain from the CoO(100) and CoO(100)-$O_V$ surfaces. Finally, the rather strong adsorption for $H_2$ (1.63 eV) at CoO(100)-$O_V$ as well as the existence of large number of active $H^{\delta-}$ species involved in this reaction can support the excellent activity of $Co_3O_4$-250.

The study of reaction pathway (Supplementary Fig. 1) shows that BHMF was the main intermediate. Therefore, kinetic studies

**Fig. 7 Hydrogenolysis of the lignin β-O-4 model compound over the Co$_3$O$_4$-250 catalyst.** Reaction conditions: substrate (0.2 g), catalyst (0.1 g), 1,4-dioxane (5 mL), H$_2$ pressure (0.5 MPa), temperature (180 °C), 8 h.

on the hydrogenolysis of BHMF were carried out. The hydrogenolysis of BHMF has a reaction order of *ca.* 0.9 for H$_2$, but close to 0 for BHMF, indicating that the activation of H$_2$ is the critical step (Supplementary Fig. 17). With D$_2$, the reaction rate is decreased by around 3.8 times (Fig. 5b), further confirming that the activation of H$_2$/D$_2$ is involved in the rate-determining step. Meanwhile, the reaction rate with D$_2$ over Co$_3$O$_4$-400 was slowed down by a factor of 2.22 comparing to that with H$_2$, due to the difference in zero-point energy between the isotopic isomers (Supplementary Fig. 18)[13].

The complete reaction pathways of the conversion of HMF to DMF over CoO(100)-O$_V$ surface have been investigated (Fig. 6 and Supplementary Fig. 19). The DFT calculations show that HMF is adsorbed on the CoO(100)-O$_V$ surface first as its adsorption is exothermic by 2.22 eV, which is higher than the adsorption of H$_2$ (exothermic of 1.63 eV) (IM1). In addition, it is adsorbed vertically on the CoO(100)-O$_v$ surface through the O atom of -CH=O, thus filling the oxygen vacancy (Supplementary Fig. 19a). Subsequently, H$_2$ is adsorbed on the CoO(100)-O$_v$ surface and this process is exothermic by 0.75 eV (IM2). The adsorbed H$_2$ is activated with a barrier of 0.60 eV in the heterolytic pathway, where one H atom is bonded to a Co atom while the other is taken by the O atom (IM2-IM3). The electronegative H of Co–H$^{\delta-}$ first attacks the electropositive C$^{\delta+}$ of the adsorbed -CH=O in HMF. This process is exothermic by 0.63 eV and needs to overcome a barrier of 0.63 eV (IM3-IM4). Otherwise, if O–H$^{\delta+}$ first attacks the O$^{\delta-}$ atom of -CH=O in HMF, it faces a higher energy barrier of 1.90 eV and is endothermic by 1.22 eV (IM3-IM1a). Therefore, these results highlight the importance of the active hydride species in the hydrogenolysis of HMF. As shown in the energy profile, the next step that the H$^{\delta+}$ attacks the O atom of -CH$_2$~O is the rate-determining step, which is endothermic by 0.97 eV and gives a barrier of 1.42 eV (IM4-IM5). The obtained intermediate BHMF is adsorbed vertically onto the CoO(100)-O$_V$ surface via the O atoms of the two terminal -CH$_2$–OH (Supplementary Fig. 19k). Then, in the process of BHMF hydrogenolysis to generate HMMF and H$_2$O, the energy barriers of 0.84 eV and 1.3 eV need to be overcome (IM7-IM9), respectively. Finally, in the process of HMMF hydrogenolysis to generate DMF and H$_2$O, the energy barriers of 1.23 and 0.33 eV need to be overcome (IM11-IM13), respectively. And the activation of C–O bonds in HMMF over Co$_3$O$_4$-250 was also demonstrated by Fourier Transform-Infrared (FTIR) spectroscopy (Supplementary Fig. 20). In addition, these mechanistic results were consistent with the experimental data (Table 1). Due to the thinner CoO films and the presence of metallic Co on the surface, Co$_3$O$_4$-300 and Co$_3$O$_4$-400 catalysts were difficult to provide sufficient and stable amounts of active H$^{\delta-}$ species and overcome the high energy barrier for the next step of hydrogenolysis reaction. As a result, Co$_3$O$_4$-300 and Co$_3$O$_4$-400 catalysts showed poor activity than that of Co$_3$O$_4$-250, with HMMF and BHMF being the main product, respectively.

**Hydrodeoxygenation of a lignin model compound over Co$_3$O$_4$-250.** To further verify the applicability of Co$_3$O$_4$-250 for the hydrodeoxygenation (HDO) of more robust biomass-derived

feedstocks, the typical lignin β-O-4 model compound was tested (Fig. 7). Complete conversion of lignin β-O-4 model compound was observed at 180 °C for 8 h with high carbon yields, affording 85.5% ethylbenzene and 84.5% cyclohexanol. Significantly, this is the first report of the reductive cleavage of β-O-4 linkage over metal oxides. In contrast, other Co$_3$O$_4$ catalysts with varying reduction temperatures all show poor activity for HDO of the lignin β-O-4 model compound (Supplementary Table 4). This result further confirms that Co$_3$O$_4$-250 has an excellent activity for the hydrogenation of C=C/C=O bonds and hydrogenolysis of C–O bonds in biomass-derived substrates. The influence of solvents on catalytic activities and product distributions were also tentatively studied through DFT calculations, and it is found that 1,4-dioxane may give the best performance among all investigated solvents (Supplementary Table 5).

## Discussion

A core-shell structured Co@CoO catalyst (Co$_3$O$_4$-250) with oxygen vacancies in the shell of CoO showed an excellent activity in the hydrogenolysis of HMF to DMF. Co$_3$O$_4$-250 exhibited a high DMF yield of 89% at 130 °C for 2 h, and the productivity is highest among all state-of-the-art catalysts to date. Co$_3$O$_4$-250 shows an excellent catalytic stability for over 100 h without notable deactivation at a high WHSV of 26.6 h$^{-1}$. The superior activity of the Co@CoO catalyst originates from the unique CoO species with suitable oxygen vacancies, which can strongly adsorb HMF and catalyse the homolytic/heterolytic splitting of H$_2$ molecules to produce highly active H$^{\delta-}$ species. This study will inspire the design of new metal-free catalysts based upon metal oxides for the hydrogenation and hydrogenolysis reactions.

## Methods

**Catalyst preparation.** Co$_3$O$_4$ was synthesized with a precipitation method. Cobalt nitrate is used as the synthetic precursor. In the typical process, 60 mmol of cobalt nitrate and 69 mmol of (NH$_4$)$_2$CO$_3$ was dissolved in 200 ml distilled water, respectively. Then the solution of (NH$_4$)$_2$CO$_3$ was added dropwise into the aqueous solution of cobalt salt under vigorous stirring until the pH of the mother liquid reached approximately 9. Finally, the suspension was aged at 65 °C for 1 h with stirring and then left to stand at room temperature for 12 h. After filtration and thoroughly washed with distilled water, the solid product was dried at 100 °C for 12 h and then calcined in air at 450 °C for 4 h to obtain Co$_3$O$_4$.

The as-prepared Co$_3$O$_4$ was further reduced at desired temperature (200–400 °C) for 2 h under flowing H$_2$ (10% H$_2$-Ar mixed gas) in tube furnace with a ramp of 5 °C·min$^{-1}$ before use. Thus-obtained reduced cobalt oxides were marked as Co$_3$O$_4$-n, in which n represents the reduction temperature.

**Catalyst activity tests in batch reactor.** Hydrogenolysis of HMF was carried out in a Teflon-lined stainless-steel autoclave (50 mL). After sealing the desired dosage of HMF, catalyst and solvent in the reactor, the autoclave was purged with H$_2$ three times to remove air and charged with the desired H$_2$ pressure. Then, the autoclave was heated to the predetermined temperature in a short time. After the reaction, the reactor was quenched in an ice-water bath immediately. The liquid phase was separated from the solid catalyst by centrifugation, and two individual GC/GC-MS systems were used for product analyses. The qualitative analysis of products was carried out on a GC-MS system (Agilent 7890A-5975C), and the quantitative analysis was executed on a GC system (Agilent 7890B) equipped with an HP-5 column and an FID detector.

**Catalyst characterization.** The electron paramagnetic resonance (EPR) spectra were collected on a Bruker A300 spectrometer at 77 K.

The Co K-edge XAS spectra were recorded at the 11B beamline at the Shanghai Synchrotron Radiation Facility in transmission and fluorescence modes. Co foil and Co oxides were used for energy calibration and as references for comparison. Athena was used for XAS data analysis, including energy calibration and spectral normalization.

Inelastic neutron scattering (INS) spectra were recorded on the VISION spectrometer at the Spallation Neutron Source, Oak Ridge National Laboratory (USA). VISION is an indirect geometry crystal analyser instrument that provides a wide dynamic range with high resolution. All the INS spectra were collected after the sample was cooled and stabilised at temperatures below 10 K. In a typical experiment, the catalyst CoO (~10 g) was loaded into a flow-type stainless-steel cell that can also be used as a static cell with all valves closed. The sample was heated at 250 °C (5 °C/min ramping) under dry He for 1 h to remove any remaining trace water before the experiment. An INS spectrum was collected upon cooling to <10 K. The sample was warmed to 130 °C (5 °C/min ramping) in a flow of $H_2$ to conduct the hydrogenation under a flow of $H_2$ for 1 h. The cell was then quenched in liquid $N_2$ and flushed briefly with dry He to remove any remaining $H_2$ in the cell and weakly adsorbed $H_2$ on the catalyst. An INS spectrum was collected upon cooling to <10 K. The sample was then heated to 250 °C (5 °C/min ramping) under a flow of $H_2$ to achieve the reduction of CoO to Co for 3.5 h. The cell was then flushed with dry He and an INS spectrum was collected upon cooling to <10 K. The sample was warmed to 130 °C (5 °C/min ramping) in a flow of $H_2$ to conduct the hydrogenation under a flow of $H_2$ for 1 h. The cell was then quenched in liquid $N_2$ and flushed briefly with dry He to remove any remaining $H_2$ in the cell and weakly adsorbed $H_2$ on the catalyst. An INS spectrum was collected upon cooling to <10 K.

**Density functional theory studies.** In this work, all spin-polarized DFT calculations were carried out using the Vienna Ab-initio Simulation Package (VASP)[53]. The projector augmented wave (PAW) method[54] and the Perdew−Burke−Ernzerhof (PBE)[55] functional under the generalized gradient approximation (GGA)[56] were applied throughout the calculations. The kinetic energy cut-off was set to 400 eV, and the force threshold in structure optimization was 0.05 eV/Å. We used a large vacuum gap of 15 Å to eliminate the interactions between neighbouring slabs. By adopting these calculation settings, the optimized lattice constant of CoO (P1) is 4.248 Å, which is in good agreement with the experimental value of 4.267 Å[57].

The transition states (TS) of surface reactions were located using a constrained optimization scheme and were verified when (i) all forces on the relaxed atoms vanish and (ii) the total energy is a maximum along the reaction coordination but it is a minimum with respect to the rest of the degrees of freedom[58–60]. The adsorption energy of species X on the surface ($E_{ads}(X)$) was calculated with

$$E_{ads}(X) = -(E_{X/slab} - E_{slab} - E_X) \qquad (1)$$

where $E_{X/slab}$ is the calculated total energy of the adsorption system, while $E_{slab}$ and $E_X$ are calculated energies of the clean surface and the gas-phase molecule X, respectively. Obviously, a positive value of $E_{ads}(X)$ indicates an exothermic adsorption process, and the more positive the $E_{ads}(X)$ is, the more strongly the adsorbate X binds to the surface.

The oxygen vacancy formation energy ($E_{OV}$) was calculated according to

$$E_{OV} = E_{slab-OV} + 1/2E_{O2} - E_{slab} \qquad (2)$$

where $E_{slab-OV}$ is the total energy of the surface with one oxygen vacancy, and $E_{O2}$ is the energy of a gas-phase $O_2$ molecule.

For the model construction, we built a $p(2 \times 3)$ surface slab containing five atomic layers for the CoO(100) surface ($a = 12.74$ Å; $b = 8.67$ Å; $c = 23.50$ Å; $\alpha = \beta = \gamma = 90°$), and the top four CoO layers were allowed to relax, while the bottom atomic layer was kept fixed to mimic the bulk region. A $2 \times 2 \times 1$ $k$-point mesh was used in calculations of all these models. Note that the on-site Coulomb interaction correction is necessary for the appropriate description of the Co $3d$ electrons, and all calculations are performed with $U = 5.1$ eV and $J = 1.0$ eV, which are consistent with the values determined by previous studies[61,62].

In addition, we tested the effect of the spin state of $3d$ electrons in $Co^{2+}$ in the optimization of CoO, and found that the high-spin antiferromagnetic arrangement was the most stable state, and the calculated magnetic moment of 2.74 μB obtained from the difference in spin-up and spin-down densities is consistent with literature reports[63–65].

## Data availability
The data supporting the findings of this study are available within the article, or available from the authors upon reasonable request.

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

## Acknowledgements

This project was supported financially by the National Natural Science Foundation of China (No. 21832002, 21825301, 21872050, 21808063, 22002043), Shanghai Municipal Science and Technology Major Project (Grant No.2018SHZDZX03), the Programme of Introducing Talents of Discipline to Universities (B16017), China and EPSRC (EP/V056409), UK. This research used Beamline VISION at the Spallation Neutron Source, a DOE Office of Science User Facility operated by the Oak Ridge National Laboratory.

## Author contributions

S.X., L.D., X.H.L. and Y.G.: preparation and characterization of catalysts, and performing the catalytic reactions. Z.Q.W. and X.Q.G.: DFT calculations. X.H., L.L.D., Y.C., A.J.R.-C. and S.Y.: collection and analysis of neutron scattering data. Q.L. and Y.F.H.: collection and analysis of XAS. S.Y., X.Q.G. and Y.Q.W.: overall direction of the project. S.X., L.D., S.Y. and Y.Q.W. wrote the manuscript with the help from all authors.

## Competing interests

The authors declare no competing interests.
