## [Peer Review File · Nature Communications]

Title: Co@CoO: a Unique Catalyst for Hydrogenolysis of Biomass-derived 5-Hydroxymethylfurfural to 2,5-DimethylfuranREVIEWER COMMENTS

Reviewer #1 (Remarks to the Author):

The Manuscript " Co@CoO: a Unique Catalyst for Hydrogenolysis of Biomass-derived 5-Hydroxymethylfurfural to 2,5-Dimethylfuran" by Yanqin Wang et al. is quite an interesting one, however, few of the points may be clarified for the benefit of the readers. My views are appended below.

1. In abstract authors claimed that this the first report on metal oxide catalyst for bio derived fuels, I would suggest to recheck the statement.
2. Authors claimed that the biotransformation occurs over partially reduced Co₃O₄, and in the form of metallic core and oxide surface, I would suggest authors may perform XAS and temperature program studies.
3. The results with and CoO and reduced CoO(Co?) are compared with partially reduced Co₃O₄ very different why!
4. The mechanism predicted though INS studies using by reference 47, how it is logical , can authors put some light on it.
5. In theoretical part, why authors have only considered CoO(111) surface ignoring the contribution from metallic Co. Since commercial CoO, did not observed any conversion of reactants.

Reviewer #2 (Remarks to the Author):

This manuscript written by Yanqin Wang, Xue-Qing Gong, et al. synthesized Co@CoO coreshell catalysts for the 5-hydroxymethylfurfural to 2,5-dimethylfuran reaction. The synthesized core-shell catalyst, namely Co₃O₄-250, revealed the superior activity among metal-oxide catalysts. Many techniques were used in experimental part to prove that the oxygen vacancies on the CoO shell promote the catalytic activity. The DFT calculations were applied to clarify the homolytic and heterolytic cleavage of H₂ over the vacancy site in CoO. After reviewing this paper, I recommend that this paper is possible to be published in Nature Communications subjected to major revision. Some parts need to be clarified. My comments are listed below.

1. In table 1, the reduced commercial CoO provided the 44.6% yield of BHMF and 1.1% yield of DMF. According to those results, the heterolytic and homolytic cleavage of H₂ around the vacancy sites of CoO may not be the only reason for the superior activity of Co@CoO. The confinement effect, morphologies, active facets as well as charge transfer between an inner core and an outer shell should influence the activity of the unique Co@CoO core-shell too. In addition, the slab models in the DFT part can represent the commercial CoO catalyst but they might not represent the Co@CoO catalyst.
2. In Fig. 2a and 2b, CoO(111), CoO (220) facets and Co(111) were observed in Co₃O₄-250. Why was CoO(100) studied in the DFT part? Typically, different facets exhibit different catalytic activity. The CoO(111) and CoO(220) facets should be studied to understand the properties of the synthesized

catalyst. In addition, the labels in Fig. 2a and 2b are too small.

3. The authors state that “The hydrogenolysis of HMMF has a reaction order of ca. 0.9 for H₂, but close to 0 for HMMF, indicating that the activation of H₂ instead of HMMF is the rate-determining step (Supplementary Figure 9), in agreement with the DFT analysis.”. However, the DFT results in the manuscript cannot conclude that the H₂ activation is the rate determining step. The complete reaction pathways of the HMF to DFF reaction should be investigated to confirm the rate-determining step of the reaction.

4. In Fig. 3a, the y-axis of the energy profile should change from energy to relative energy. Is the -E_{ads} of HMF on CoO(100)-Ov, -1.36eV, wrong? There are only two data points for the E_{ads} of HMF and they are not relevant to the energy profile of H₂. They should be removed from Fig. 3 and describing those values in the text would be better.

5. More configurations of HMF adsorption should be tested. For example, the bridge adsorption configuration might be formed via hydroxyl and formyl groups binding with a surface proposed by Kim et al. [Applied Surface Science, 456, 2018, 174-183] The Bader charges of atoms in HMF on catalyst should be analyzed.

6. In the DFT method, the information of a CoO crystal (i.e. space group) and size of a slab model, in Å, should be given.

Reviewer #3 (Remarks to the Author):

This contribution of the Wang team is an interesting work toward the better understanding of catalyst design for desired biomass deoxygenation reactions, in particular C-O hydrogenolysis, C=O hydrogenation as well as hydrodeoxygenation. Despite the relative simplicity of the substrates, this work addresses a central question, important in this field now for decades: how to increase efficiency for desired selective deoxygenation processes by more rational catalyst design. The team has found a unique core shell type Cobalt containing catalyst with activity outperforming even noble metal catalysts for the same reaction. Here the Co-oxide has been involved in the hydrogen activation, which as an important finding.

I recommend the paper to be published after the questions below are addressed:

1.) It is known in the literature that conversion of HMF to DMF goes through many intermediates, and is known for side reactions. The side reactions are mainly char formation through self-condensation, undesired ring opening and subsequent condensation – these lead to loss of mass balance. Another set of side reactions are the ring hydrogenation products as well as ring opening and consecutive hydrogenation products. The reaction intermediates involve the corresponding furfural derivatives, and derived hydrogenation and hydrogenolysis products that are all part of the reaction network. There are typically more intermediates compared to what is shown on the scheme in Table 1. It would be great to comment on these aspects, contrast to or acknowledge already existing literature findings, also with references either in main text or in supporting.

Have the authors not observed these phenomena? The recondensation is trackable by strictly following

the mass balance. Were there no ring hydrogenation intermediates? This is surprising given the fact that the lignin b-O-4 model compound products include cyclohexanol. Data on internal standard/ mass balance and GC-MS/FID (representative) figures should be provided in the supporting information.

2.) Figure 1a: The comparison with existing systems is appreciated and appears to be complete. Can the area in the graph be somewhat enlarged, since it is a bit difficult to read. Also it appears that system 14 and system 19 are missing on the graph.

3.) It has been earlier observed in the literature that the 5-HMF hydrogenation is solvent dependent. Some prior art has been demonstrated in alcohols as solvents and have shown superior activities. In this work THF was used for the 5-HMF conversion and dioxane for the lignin b-O-4 conversion. Regarding this:

a.) for future application, and given that the topic of the paper focuses on sustainability and moving from noble metal catalysts, do the authors think that such solvent choice is suitable?

b.) a solvent evaluation table would be useful to add in supporting information

c.) with such high hydrogenolysis activity of the catalyst why the choice of the ethers as solvents? Did they not suffer hydrogenolysis? A blank reaction with these solvents along should be added to the table.

d.) Are these solvents innocent in catalysis or did they play any role in interacting with the catalyst surface, potentially coordinating to active metal sites etc..? And these aspects should be commented if possible supported by data or calculation.

4.) In the conclusions, the authors state: 'The superior activity of the Co@CoO catalyst originates from the unique CoO species with suitable oxygen vacancies, which can strongly adsorb HMF and catalyze the homolytic/heterolytic splitting of H₂ molecules'. I assume they mean strong C=O interaction.

In my view there was an extensive study on the H₂ activation with this catalyst, but relatively less information was given related to the substrate – to – catalyst interactions. Can the authors measure some of these aspects? And especially aspects related to hydrogenolysis. It is also stated earlier in the manuscript that HMF interacts strongly with C=O and is rapidly hydrogenated (which may explain the lack of self-condensation) but perhaps the even more interesting question is the hydrogenolysis step. How does a C-O bond interact with the catalyst? Is there an interaction with the aromatic rings with this catalyst? (furan as well as phenol derivatives). The authors should provide more details on these aspects, experimental or calculation, to strengthen the narrative of the paper.

Responses to the reviewers' comments

Reviewer #1:

The Manuscript " Co@CoO: a Unique Catalyst for Hydrogenolysis of Biomass-derived 5-Hydroxymethylfurfural to 2,5-Dimethylfuran" by Yanqin Wang et al. is quite an interesting one, however, few of the points may be clarified for the benefit of the readers. My views are appended below.

1. In abstract authors claimed that this is the first report on metal oxide catalyst for bio derived fuels, I would suggest to recheck the statement.

Reply: The statement has been revised: "This study unlocks, for the first time, the potential of simple metal-oxide-based catalysts for the hydrodeoxygenation of renewable biomass to chemical feedstocks."

2. Authors claimed that the biotransformation occurs over partially reduced Co_3O_4 , and in the form of metallic core and oxide surface, I would suggest authors may perform XAS and temperature program studies.

Reply: The Co K-edge XAS spectra of Co_3O_4 -200, Co_3O_4 -250 and Co_3O_4 -300 have been measured and the spectra in R space are shown in Fig. 1 with Co foil, CoO and Co_3O_4 as references. The local environment of Co in Co_3O_4 -250 and Co_3O_4 -300 is similar to that of Co foil^{1,2} with a dominant Co-Co feature at 2.18 Å. Compared with the Co foil, the amplitude of this feature is much lower for Co_3O_4 -250 and Co_3O_4 -300, indicating that the Co species in these two catalysts are not as well ordered as metallic Co. Meanwhile, there is no metallic Co-Co feature at 2.18 Å for Co_3O_4 -200, which is similar to that of CoO with peaks at around 1.45 and 2.60 Å, corresponding to Co-O and Co-Co distances, respectively. Additionally, a weak signal at 2.70 Å is observed for Co_3O_4 -250, similar to the Co-O distance in the CoO standard, further confirming that the sample is not reduced completely and there is still a small amount of CoO species. These results indicate that the Co species in Co_3O_4 -200 is CoO, and the metallic Co and CoO species co-exist on Co_3O_4 -250, which are in agreement with the HRTEM results. This discussion has been added in the revised manuscript (main text and Supplementary Figure 9).

Fig. 1. The Co K-edge XAS spectra in R space of Co_3O_4 -200, Co_3O_4 -250, Co_3O_4 -300, Co foil, CoO and Co_3O_4 , respectively.

H_2 -TPR profile of Co_3O_4 sample is shown in Fig. 2. Two broad reduction peaks appear at 285 °C and 392 °C. The former can be attributed to the reduction of Co_3O_4 to CoO, while the reduction of CoO to metallic Co results in the latter. Therefore, the Co_3O_4 -250 (reduction at 250 °C for 2h) sample is not reduced completely.

Fig. 2. H_2 -TPR profile of Co_3O_4 sample.

- 1) Lukashuk, L., Föttinger, K., Kolar, E., Rameshan, C., Teschner, D., Hävecker, M., Knop-Gericke, A., Yigit, N., Li, H., McDermott, E., Stöger-Pollach, M. & Rupprechter, G. Operando XAS and NAP-XPS studies of preferential CO oxidation on Co_3O_4 and CeO_2 - Co_3O_4 catalysts, *J. Catal.* **344**, 1-15, (2016).

2) Zhang, L., Shang, N., Gao, S., Wang, J., Meng, T., Du, C., Shen, T., Huang, J., Wu, Q., Wang, H., Qiao, Y., Wang, C., Gao, Y. & Wang, Z. Atomically dispersed Co catalyst for efficient hydrodeoxygenation of lignin-derived species and hydrogenation of nitroaromatics. *ACS Catal.* **10**, 8672-8682, (2020).

3. The results with CoO and reduced CoO (Co⁰) are compared with partially reduced Co₃O₄ very different why!

Reply: We thank the reviewer for raising this interesting question. The catalytic performance can be significantly influenced by the particle size, morphology, and surface properties of the catalysts. For the catalysts studied in this work, the commercial CoO is a bulky material with low surface area ($12 \text{ m}^2 \cdot \text{g}^{-1}$), while Co₃O₄ has a high surface area ($30 \text{ m}^2 \cdot \text{g}^{-1}$). More importantly, reduced commercial CoO appear like large particles (Fig. 3), while Co₃O₄-250 is a core-shell structured catalyst and the CoO shell is very thin, ca. 2.5-3.5 nm, which may contribute positively to its catalytic activity (e.g., through the quantum size effect³⁻⁵). The detailed influence of CoO particle size and morphology may require future extensive investigations, which we aim to publish separately. This work mainly focused on the study of the CoO shell with oxygen vacancies that can dissociate H₂ and activate HMF, giving rise to high activity and stability.

Fig. 3. The TEM images of commercial CoO reduced at 250 °C.

3) Ma, L.-Y., Tang, L., Guan, Z.-L., He, K., An, K., Ma, X.-C., Jia, J.-F., Xue, Q.-K., Han, Y., Huang, S., Liu, F. Quantum size effect on adatom surface diffusion. *Phys. Rev. Lett.* **97**, 266102 (2006)

4) Song, C.-L., Wang, Y.-L., Ning, Y.-X., Jia J.-F., Chen, X., Sun, B., Zhang, P., Xue, Q.-K., Ma, X. C. Tailoring phthalocyanine metalation reaction by quantum size effect. *J. Am. Chem. Soc.* **132(5)**, 1456–1457 (2010)

5) Saha, S., Kuamr, J. S., Murmu, N. C., Samanta, P., Kuila, T. Controlled electrodeposition of iron oxide/nickel oxide@Ni for the investigation of the effects of stoichiometry and particle size on energy storage and water splitting applications. *J. Mater. Chem. A* **6**, 9657-9664 (2018)

4. The mechanism predicted though INS studies using by reference 47, how it is logical, can authors put some light on it.

Reply: We thank the reviewer for the advice. The sentence has been revised: “A broad underlining intensity across 80-160 meV is also observed, and this can be assigned to the formation of Co-O-H moieties; a similar feature at 75-150 meV has been observed previously upon the formation of Fe-O-H during the heterolytic dissociation of H₂ over a CuCrFeO_x catalyst”.

During the heterolytic dissociation of H₂, the direct capture of hydride (H⁻) and hydroxyl (H⁺) is important but difficult. INS study in reference 47 gave the direct evidence: the broad signal at 75-150 meV is attributed to the formation of Fe-O-H (H⁺). In our INS spectra, the main feature centred at 110 meV evidently indicates the formation of Co-H species. Meanwhile, a broad intensity across 80-160 meV is also observed, and this can be assigned to the formation of Co-O-H. Therefore, Co-H hydride and Co-O-H species coexisted on the surface during H₂ activation.

5. In theoretical part, why authors have only considered CoO(111) surface ignoring the contribution from metallic Co. Since commercial CoO, did not observed any conversion of reactants.

Reply: We thank the reviewer for raising this concern. The HRTEM result suggests that the thickness of the CoO shell is 2.5 to 3.5 nm. In addition, as shown in Table 1, compared with

the Co@CoO (Co₃O₄-250) catalyst (89.2% yield of DMF), Co₃O₄-400 that is mainly a metallic Co catalyst has very poor activity (1.3% yield of DMF), indicating that it is the CoO that results in the high activity. Furthermore, a combination of XPS, EPR and INS analyses also suggest that the shell of CoO with O_v is the main active phase. Thus, we sought to reveal the unique activity of the CoO-O_v site in this work, and the core of Co was not included in the theoretical modelling.

Reviewer #2:

This manuscript written by Yanqin Wang, Xue-Qing Gong, et al. synthesized Co@CoO core-shell catalysts for the 5-hydroxymethylfurfural to 2,5-dimethylfuran reaction. The synthesized core-shell catalyst, namely Co₃O₄-250, revealed the superior activity among metal-oxide catalysts. Many techniques were used in experimental part to prove that the oxygen vacancies on the CoO shell promote the catalytic activity. The DFT calculations were applied to clarify the homolytic and heterolytic cleavage of H₂ over the vacancy site in CoO. After reviewing this paper, I recommend that this paper is possible to be published in Nature Communications subjected to major revision. Some parts need to be clarified. My comments are listed below.

1. In table 1, the reduced commercial CoO provided the 45.0% yield of BHMF and 1.1% yield of DMF. According to those results, the heterolytic and homolytic cleavage of H₂ around the vacancy sites of CoO may not be the only reason for the superior activity of Co@CoO. The confinement effect, morphologies, active facets as well as charge transfer between an inner core and an outer shell should influence the activity of the unique Co@CoO core-shell too. In addition, the slab models in the DFT part can represent the commercial CoO catalyst but they might not represent the Co@CoO catalyst.

Reply: We thank the reviewer for these valuable comments. We fully agree with the reviewer that the particle size, morphology, surface properties and confinement effect can affect the activity of the catalyst. In this work, Co₃O₄-250 shows better catalytic performance than the reduced commercial CoO catalyst, and this could be (partially) originated from the CoO shell with oxygen vacancies of the former. The thickness of the CoO shell is very thin, ca. 2.5-3.5 nm, which may contribute to the exceptional activity (e.g., through quantum size effect¹⁻³).

Also, the influence of morphology and charge transfer may be non-trivial. This will deserve a full future investigation. Honestly, we focused on exploring the unique role of the CoO species with suitable oxygen vacancies and revealed that such active sites can strongly adsorb HMF, and also catalyze the homolytic/heterolytic splitting of H₂ molecules to produce highly active H^{δ-} species. Please see point #5 of reviewer 1 for the response to the DFT modelling.

1) Ma, L.-Y., Tang, L., Guan, Z.-L., He, K., An, K., Ma, X.-C., Jia, J.-F., Xue, Q.-K., Han, Y., Huang, S., Liu, F. Quantum size effect on adatom surface diffusion. *Phys. Rev. Lett.* **97**, 266102 (2006)

2) Song, C.-L., Wang, Y.-L., Ning, Y.-X., Jia J.-F., Chen, X., Sun, B., Zhang, P., Xue, Q.-K., Ma, X. C. Tailoring phthalocyanine metalation reaction by quantum size effect. *J. Am. Chem. Soc.* **132(5)**, 1456–1457 (2010)

3) Saha, S., Kuamr, J. S., Murmu, N. C., Samanta, P., Kuila, T. Controlled electrodeposition of iron oxide/nickel oxide@Ni for the investigation of the effects of stoichiometry and particle size on energy storage and water splitting applications. *J. Mater. Chem. A* **6**, 9657-9664 (2018)

2. In Fig. 2a and 2b, CoO(111), CoO (220) facets and Co(111) were observed in Co₃O₄-250. Why was CoO(100) studied in the DFT part? Typically, different facets exhibit different catalytic activity. The CoO(111) and CoO(220) facets should be studied to understand the properties of the synthesized catalyst. In addition, the labels in Fig. 2a and 2b are too small.

Reply: We thank the reviewer for raising these concerns. The reasons for choosing CoO (100) in the DFT part are: (a) The HRTEM results showed that the CoO (200) facet was observed together with the CoO (111) and CoO (220) facets at Co₃O₄-250. (b) Compared with the CoO (111) facet, CoO (100) is thermodynamically more stable. Nevertheless, we have calculated the energy profiles of the adsorption and dissociation of H₂ on the CoO(111) and CoO(111)-O_v surfaces, and the results are presented in Fig. 3 and Fig. 4, respectively. For the homolytic dissociation of H₂, it is exothermic by 3.32 eV and gives a barrier of 0.91 eV at CoO(111). On the CoO(111)-O_v surface, it is also exothermic (3.56 eV), but the barrier is significantly reduced to 0.32 eV. This result indicates that CoO(111)-O_v is more beneficial than CoO(111) for H₂ dissociation, in agreement with the experimental measurement of the CoO(100) and

CoO(100)-O_v surfaces. Notably, the CoO(111) surface is a polar surface with O-terminated structure, which tends to form stable OH species. Even though, H₂ on the CoO(111)-O_v surface can be still split into one H^{δ+} and one H^{δ-} with the help of the O_v, as confirmed by the Bader charge analysis (Fig. 4), and such heterolytic dissociation of H₂ was calculated to be exothermic (1.59 eV) and with an energy barrier of 0.41 eV. In comparison, the CoO(100)-O_v surface is more favourable to produce a large number of active H^{δ-} species due to the strong adsorption for H₂ (1.63 eV), and the H^{δ-} species can be also produced in a homolytic way (Fig. 7). All these results show that the hydride can be produced by H₂ dissociation on both the reduced CoO(111) and Co(100) surfaces. The related discussions are now added in the revised manuscript (main text and Supplementary Figures 14 and 15).

Fig. 3. (a) Calculated energy profiles of adsorption and dissociation of H₂ on the CoO(111) and CoO(111)-O_v surfaces. H₂(g): gas-phase H₂; H₂*: adsorbed H₂ on surface; H₂(TS): the adsorbed H₂ on surface dissociates to two adsorbed H on surface; 2H*: the co-adsorption of two H on surface.

Fig. 4. Calculated adsorption energies, structures and Bader charges of H₂ dissociation on (a)-(d) CoO(111) surface and (e)-(i) CoO(111)-O_v surface.

3. The authors state that “The hydrogenolysis of BHMF has a reaction order of ca. 0.9 for H₂, but close to 0 for BHMF, indicating that the activation of H₂ instead of BHMF is the rate-determining step (Supplementary Figure 9), in agreement with the DFT analysis.”. However, the DFT results in the manuscript cannot conclude that the H₂ activation is the rate determining step. The complete reaction pathways of the HMF to DMF reaction should be investigated to confirm the rate-determining step of the reaction.

Reply: We thank the reviewer for this suggestion. We have conducted new calculations and revised the statement: “The hydrogenolysis of BHMF has a reaction order of *ca.* 0.9 for H₂, but close to 0 for BHMF, indicating that the activation of H₂ is a critical step (see Supplementary Figure 17)”

The complete reaction pathways of HMF to DMF on the CoO(100)-O_v surface were investigated (Fig. 5 and Fig. 6). The DFT calculations showed that HMF is adsorbed on the CoO(100)-O_v surface first as its adsorption is exothermic by about 2.22 eV, which is higher than the adsorption of H₂ (exothermic of 1.63 eV). In addition, it is adsorbed vertically on the

CoO(100)-O_v surface through the O atom of -CH=O by filling the oxygen vacancy. Subsequently, H₂ is adsorbed on the CoO(100)-O_v surface and this process is exothermic by 0.75 eV. The adsorbed H₂ is activated with a barrier of 0.60 eV in the heterolytic way, where one H atom is bonded with Co atom while the other is taken by the O atom. The electronegative H of Co-H^{δ-} first attacks the electropositive C^{δ+} of the adsorbed -CH=O in HMF. This process is exothermic by 0.63 eV and needs to overcome a barrier of 0.63 eV. Otherwise, if O-H^{δ+} first attacks the O^{δ-} atom of -CH=O in HMF, this process needs a higher energy barrier of 1.90 eV and is endothermic by 1.22 eV. Therefore, these results highlight the importance of the active hydride species in the hydrogenolysis reaction of HMF. As shown in the energy profiles, the next step that the H^{δ+} attacks the O atom of -CH₂-O is the rate determining step, which is endothermic by 0.97 eV and gives a barrier of 1.42 eV. The obtained intermediate BHMF is adsorbed vertically onto the CoO(100)-O_v surface via the O atoms of the two terminal -CH₂-OH. Then, in the process of BHMF hydrogenolysis to generate HMMF and H₂O, the energy barriers of 0.84 eV and 1.3 eV need to be overcome, respectively. Finally, in the process of HMMF hydrogenolysis to generate DMF and H₂O, the energy barriers of 1.23 eV and 0.33 eV need to be overcome, respectively. This discussion was added in the revised manuscript (main text, Fig. 6 and Supplementary Figure 19).

Fig. 5. Calculated energy profiles of HMF hydrogenolysis reaction on the CoO(100)-O_v surface. Red dotted line refers to the H⁺ route, black dotted line is the H⁻ route.

Fig. 6. Calculated structures and adsorption energies of the key species within HMF hydrogenolysis reaction on CoO(100)-O_v surface.

4. In Fig. 3a, the y-axis of the energy profile should change from energy to relative energy. Is the -E_{ads} of HMF on CoO(100)-O_v, -1.36eV, wrong? There are only two data points for the E_{ads} of HMF and they are not relevant to the energy profile of H₂. They should be removed from Fig. 3 and describing those values in the text would be better.

Reply: We thank the reviewer for raising this concern. The adsorption energy of HMF (E_{ads}) at CoO(100)-O_v is exothermic by about 2.22 eV (Fig.8 (g)). As shown in Fig. 7, the two data points for the adsorption energy of HMF (E_{ads}) have been removed. Please see our reply to

comment 5 below for a detailed explanation.

Fig. 7. Calculated energy profiles of adsorption and dissociation of H₂ on the CoO(100) and CoO(100)-O_v surfaces. H₂(g): gas-phase H₂; H₂*: adsorbed H₂ on surface; H₂(TS): the adsorbed H₂ on surface dissociates to two adsorbed H on surface; 2H*: the co-adsorption of two H on surface.

5. More configurations of HMF adsorption should be tested. For example, the bridge adsorption configuration might be formed via hydroxyl and formyl groups binding with a surface proposed by Kim et al. [Applied Surface Science, 456, 2018, 174-183] The Bader charges of atoms in HMF on catalyst should be analyzed.

Reply: We thank the reviewer for this valuable suggestion. Inspired by this reference, we have included additional HMF adsorption modes at CoO(100) and CoO(100)-O_v in the calculation (Fig. 8):

(a) HMF can be parallelly and vertically adsorbed on the two surfaces via C=O or C-OH groups. The two possible bridging adsorption structures were determined and the calculated adsorption energies at CoO(100) are 0.71 eV and 0.47 eV (Fig. 8(a) and (b)), respectively; while at CoO(100)-O_v, the adsorption energies are 1.03 eV and 2.14 eV (Fig. 8(l) and (h)), respectively.

(b) The calculated adsorption energies at CoO(100)-O_v are higher than those at CoO(100) surface, indicating that the CoO(100)-O_v possesses a stronger binding affinity.

(c) The calculated highest adsorption energy of HMF on the CoO(100) surface is only

0.71 eV, in agreement with the poor activity of the commercial CoO catalyst, and these results are consistent with the above literature study.

(d) The adsorption energy determined for the interaction of HMF with the surface through the C=O group is the strongest (2.22 eV) among all those determined on the CoO(100)-O_v surface, which is consistent with the improved performance of the first hydrogenation step of HMF to BHMF, followed by hydrogenolysis to HMMF and DMF at the Co₃O₄-250 catalyst.

In addition, we also calculated the Bader charges of the atoms in the adsorbed HMF on the CoO(100) and CoO(100)-O_v surfaces. The results showed that the stronger the HMF adsorption is, the more electrons the HMF can obtain from the CoO(100) and CoO(100)-O_v surfaces. **The extended HMF adsorption modes and the corresponding discussion have been added in the revised manuscript (main text and Supplementary Figure 16).**

Fig. 8. Calculated adsorption structures, energies, and Bader charges of HMF (left: top view, right: side view) at (a)–(f) CoO(100) (red background) and (g)–(l) CoO(100)-O_v (yellow background).

6. In the DFT method, the information of a CoO crystal (i.e. space group) and size of a slab model, in Å, should be given.

Reply: We thank the reviewer for this suggestion. We have modified this section in the revised manuscript. The optimized lattice constant of CoO (P1) is 4.248 Å, which is in good agreement

with the experimental value of 4.267 Å¹.

For the model construction, we built a $p(2\times 3)$ surface slab containing five atomic layers for the CoO(100) surface ($a = 12.74$ Å; $b = 8.67$ Å; $c = 23.50$ Å; $\alpha = \beta = \gamma = 90^\circ$), and the top four CoO layers were allowed to relax, while the bottom atomic layer were fixed to mimic the bulk region.

- 1) Redman, M. J. & Steward, E. G. Cobaltous oxide with the zinc blende/wurtzite-type crystal structure. *Nature* **193**, 867-867 (1962).

Reviewer #3:

This contribution of the Wang team is an interesting work toward the better understanding of catalyst design for desired biomass deoxygenation reactions, in particular C-O hydrogenolysis, C=O hydrogenation as well as hydrodeoxygenation. Despite the relative simplicity of the substrates, this work addresses a central question, important in this field now for decades: how to increase efficiency for desired selective deoxygenation processes by more rational catalyst design. The team has found a unique core shell type Cobalt containing catalyst with activity outperforming even noble metal catalysts for the same reaction. Here the Co-oxide has been involved in the hydrogen activation, which as an important finding.

I recommend the paper to be published after the questions below are addressed:

1.) It is known in the literature that conversion of HMF to DMF goes through many intermediates, and is known for side reactions. The side reactions are mainly char formation through self-condensation, undesired ring opening and subsequent condensation - these lead to loss of mass balance. Another set of side reactions are the ring hydrogenation products as well as ring opening and consecutive hydrogenation products. The reaction intermediates involve the corresponding furfural derivatives, and derived hydrogenation and hydrogenolysis products that are all part of the reaction network. There are typically more intermediates compared to what is shown on the scheme in Table 1. It would be great to comment on these aspects, contrast to or acknowledge already existing literature findings, also with references either in main text or in supporting.

Reply: We thank the reviewer for the helpful advice. The extended reaction network and corresponding discussions have been added in the revised manuscript (main text and Figure 1).

Have the authors not observed these phenomena? The recondensation is trackable by strictly

following the mass balance. Were there no ring hydrogenation intermediates? This is surprising given the fact that the lignin β -O-4 model compound products include cyclohexanol. Data on internal standard/ mass balance and GC-MS/FID (representative) figures should be provided in the supporting information.

Reply: We thank the reviewer for raising this concern. No ring hydrogenation or opening products were observed in this system. The data on internal standard/mass balance and representative GC-MS/FID figures are shown below and has been **added in the revised supporting information (Supplementary Figure 2)**. The results confirm that only DMF, HMMF and BHMF were observed.

Fig. 10. GC-MS/FID spectra of (a) Co_3O_4 -250, (b) Co_3O_4 -300 and (c) Co_3O_4 -400 in the HMF hydrogenolysis reaction.

For the hydrogenolysis of lignin β -O-4 model compound, phenol is the main intermediate in the reaction network, which is easily hydrogenated to cyclohexanol under the reaction condition^{1,2}.

- 1) Nakagawa, Y., Ishikawa, M., Tamura, M. & Tomishige, K. Selective production of cyclohexanol and methanol from guaiacol over Ru catalyst combined with MgO. *Green Chem.* **16**, 2197-2203 (2014).
- 2) Gundekari, S. & Karmee, S. K. Selective synthesis of cyclohexanol intermediates from

lignin-based phenolics and diaryl ethers using hydrogen over supported metal catalysts: a critical review. *Catal. Surv. Asia* **25**, 1-26 (2020).

2.) Figure 1a: The comparison with existing systems is appreciated and appears to be complete. Can the area in the graph be somewhat enlarged, since it is a bit difficult to read. Also it appears that system 14 and system 19 are missing on the graph.

Reply: We thank the reviewer for the kind reminder, and we have corrected these issues in the revised manuscript.

3.) It has been earlier observed in the literature that the 5-HMF hydrogenation is solvent dependent. Some prior art has been demonstrated in alcohols as solvents and have shown superior activities. In this work THF was used for the 5-HMF conversion and dioxane for the lignin b-O-4 conversion. Regarding this:

a.) for future application, and given that the topic of the paper focuses on sustainability and moving from noble metal catalysts, do the authors think that such solvent choice is suitable?

Reply: We thank the reviewer for this comment. We carried out the hydrogenation and hydrogenolysis of HMF in ethanol and isopropanol under the same reaction conditions. The results are shown in Table 1. The HMF conversion all reached 99% in 2h in these two solvents, comparable with that in THF, but the yields of DMF are lower, with HMMF as the intermediate and 5-methyl furfural(5-MF), ring hydrogenation by-products (MTHFA; DMTHF) and 2-hexanol as by-products. The existence of intermediate may be due to the co-adsorption of hydroxyl groups in alcohols with that in intermediates on catalyst surface, and thus slow down the hydrogenolysis of intermediate to DMF. The existence of byproducts may be attributed to the contribution of alcohols, which are known as good hydrogen sources. In short, these results indicate that solvent can influence the catalytic performance and THF has given the best performance. These results and discussion were added in the revised manuscript (Supplementary Table 1).

Table 1. The solvent evaluation for the HMF hydrogenolysis reaction*.

Solvent	Conv. (%)	Yield (%)				Mass balance (%)
		BHMF	HMMF	DMF	Others [#]	
THF	>99	0.0	4.3	89.2	0.0	94.5
Ethanol	>99	0.0	10.1	58.0	22.8	91.9
Isopropanol	>99	0.2	13.8	63.5	16.4	94.9

* Reaction conditions: HMF: 0.15 g, Co₃O₄-250: 0.03 g, solvent: 5 mL, H₂: 1.0 MPa, 130 °C, 2 h.

Others mainly include 5-methyl furfural (5-MF), ring hydrogenation by-products (MTHFA; DMTHF) and 2-hexanol.

The catalytic activities and product distributions for the hydrogenolysis of lignin β-O-4 model compound in different solvents have also been carried out (Table 2). It is found that among all solvents, 1,4-dioxane showed the highest catalytic performance and the yield of ethylbenzene and cyclohexanol reach 78.9% and 77.0%, respectively, with nearly full conversion of lignin model compound. A few by-products are detected, such as 1-methoxy-2-phenethoxybenzene (13.0%). When using THF as the solvent, the yield of ethylbenzene and cyclohexanol was only 66.6% and 63.3%, respectively. Meanwhile, the yield of 1-methoxy-2-phenethoxybenzene was up to 22.7%, giving lower hydrogenolysis activity than that in 1,4-dioxane. When ethanol was used as solvent, the conversion of lignin β-O-4 model compound is 94.5% and the products are predominantly benzene ring retaining products (ethylbenzene and guaiacol), which is consistent with the literature^{1,2} These results indicate that the hydrogenolysis performance of lignin β-O-4 model compound is influenced by solvents, and dioxane shows the best performance.

Table 2. The solvent evaluation for lignin β-O-4 model compound hydrogenolysis reaction*.

Solvent	Conv. (%)	Yield (%)					Mass balance (%)
		1	2	3	4	5	
1,4-Dioxane	100	78.9	77.0	--	13.0	1.2	93.1/91.2
THF	100	66.6	63.3	4.9	22.7	--	89.3/90.9
Ethanol	94.5	65.4	--	59.8	32.9	--	103.8/98.2

*Reaction conditions: β-O-4 model compound: 0.2 g, Co₃O₄-250: 0.1 g, solvent: 5 mL, H₂: 0.5 MPa, 180 °C, 4 h.

- 1) Lin, F. *et al.* Identification of the cleavage mechanisms and hydrogenation activity of the β-O-4 linkage in a lignin model compound over Ni-CeO₂/H-ZSM-5. *Appl. Catal. A: Gen.* **598**, 117552 (2020).

2) Chen, C., Liu, P., Xia, H., Zhou, M. & Jiang, J. Catalytic transfer hydrogenation of 4-O-5 models in lignin-derived compounds to cycloalkanes over Ni-based catalysts. *J.Chin. Chem. Soc.* **68**, 582-591 (2020).

b.) a solvent evaluation table would be useful to add in supporting information

Reply: The results using different solvents have been added in the revised manuscript (Supplementary Table 2).

c.) with such high hydrogenolysis activity of the catalyst why the choice of the ethers as solvents? Did they not suffer hydrogenolysis? A blank reaction with these solvents along should be added to the table.

Reply: We have carried out two reactions without precursors and no hydrogenolysis products of THF or dioxane were detected in liquid or gas phase. The GC-MS and GC spectra are shown in Fig. 11/ Fig. 12 and added in the revised manuscript (Supplementary Figures 3 and 4).

Fig. 11. (a) GC-MS spectra of liquid products, (b) GC spectra with TCD detector of gas products and (c) GC spectra with FID detector of gas products. Reaction conditions: Co_3O_4 -250: 0.03 g, THF: 5 mL, H_2 : 1.0 MPa, 130 °C, 2 h.

Fig. 12. (a) GC-MS spectra of liquid products, (b) GC spectra with TCD detector of gas products and (c) GC spectra with FID detector of gas products. Reaction conditions: Co_3O_4 -250: 0.03 g, dioxane: 5 mL, H_2 : 1.0 MPa, 180 °C, 2 h.

d.) Are these solvents innocent in catalysis or did they play any role in interacting with the catalyst surface, potentially coordinating to active metal sites etc.? And these aspects should be commented if possible supported by data or calculation.

Reply: We thank the reviewer for raising this comment. From the data and discussion given above, we find that THF is better than conventional solvents. To illustrate the reasons, we calculated the adsorptions of these solvent molecules (THF and ethanol) on the $\text{CoO}(100)\text{-O}_v$ surface (Fig. 13) and found that, (a) the calculated highest adsorption energies of THF and ethanol on the $\text{CoO}(100)\text{-O}_v$ surface are 1.27 eV and 2.16 eV, respectively, both of which are below that of HMF (2.22 eV); (b) as expected, the adsorption energy of ethanol is very close to that of HMF, and the competitive adsorption likely leads to the low yield of DMF in alcohols.

These results are now added in the revised manuscript (Supplementary Figure 5).

Fig. 13. Calculated adsorption energies and structures of (a)–(c) THF (left: top view, right: side view) at CoO(100)-O_v surface, which are marked with red background, and (d)–(f) ethanol (left: top view, right: side view) at CoO(100)-O_v surface, which are marked with blue background.

4.) In the conclusions, the authors state: ‘The superior activity of the Co@CoO catalyst originates from the unique CoO species with suitable oxygen vacancies, which can strongly adsorb HMF and catalyse the homolytic/heterolytic splitting of H₂ molecules’. I assume they mean strong C=O interaction.

In my view there was an extensive study on the H₂ activation with this catalyst, but relatively less information was given related to the substrate – to – catalyst interactions. Can the authors measure some of these aspects? And especially aspects related to hydrogenolysis. It is also stated earlier in the manuscript that HMF interacts strongly with C=O and is rapidly hydrogenated (which may explain the lack of self-condensation) but perhaps the even more interesting question is the hydrogenolysis step. How does a C-O bond interact with the catalyst? Is there an interaction with the aromatic rings with this catalyst? The authors should provide more details on these aspects, experimental or calculation, to strengthen the narrative of the paper.

Reply: We appreciate the reviewer for the suggestion. Additional DFT calculations show that C=O and C-O bonds in HMF prefer to be vertically adsorbed on the catalyst surface. Furthermore, HMMF-adsorption-IR analysis was carried out and the results are shown in Fig. 14. There is a red shift of the signal at 1023 cm⁻¹ for C-O stretching in the alkoxy functional group^{1,2}. This phenomenon indicated that Co₃O₄-250 has an activating effect on the C-O bonds. In comparison, the characteristic peaks at 1500-1650 cm⁻¹ for C=C stretching and 1198 cm⁻¹

for C-O-C stretching, which can be assigned to furan skeleton mode, show a blue shift. It suggests that the stretching vibration of C=C and C-O-C becomes stronger, which may be attributed to the activation of C-O bond in the alkoxy functional group³. **These results have been added in the revised manuscript (Supplementary Figure 20).**

Fig. 14. DRIFTS spectra of 5-methyl-2-furanmethanol (HMMF) adsorbed on Co₃O₄-250 at 30 °C followed by Ar flushing.

In addition, the complete reaction pathways and adsorption structures of HMF to DMF reaction over CoO(100)-O_v surface have also been investigated. As shown in Fig. 6, the obtained intermediate BHMF is adsorbed vertically onto the CoO(100)-O_v surface via the O atoms of the two terminal -CH₂-OH (Fig. 6k). Another intermediate, HMMF is also adsorbed vertically onto the CoO(100)-O_v surface via the O atoms of -CH₂-OH (Fig. 6q). Notably, the furan-ring of these adsorption structures have no interaction of the CoO(100)-O_v surface. More analysis and discussions have been provided in our response to the comment #3 of Reviewer 2 (Fig. 5 and Fig. 6).

- 1) Yang, H., Yan, R., Chen, H., Lee, D. H. & Zheng, C. Characteristics of hemicellulose, cellulose and lignin pyrolysis. *Fuel*. **86**, 1781-1788, (2007).
- 2) Hu, J., Zhao, M., Jiang, B., Wu, S. & Lu, P. Catalytic Transfer Hydrogenolysis of Native Lignin to Monomeric Phenols over a Ni-Pd Bimetallic Catalyst. *Energy & Fuels*. **34**, 9754-9762, (2020).
- 3) Dong, L., Xia, J., Guo, Y., Liu, X., Wang, H. & Wang, Y. Mechanisms of C_{aromatic}-C bonds cleavage in lignin over NbO_x-supported Ru catalyst. *J. Catal.* **394**, 94-103, (2021).

REVIEWERS' COMMENTS

Reviewer #1 (Remarks to the Author):

Authors have addressed all the queries raised by me satisfactorily, therefore may be considered for publication.

Reviewer #2 (Remarks to the Author):

In the revised manuscript, the authors addressed all comments properly. However, there are some mistakes in the revised manuscript and SI. After reviewing this paper, I recommend that this paper is possible to be published in Nature Communications subjected to minor revision. Additional comments are listed below.

1. Scheme in Figure 1 in the revised manuscript is wrong. There is only hydrogenation ($+H_{2}$) in the HMF-to- BHMF step.
2. The authors conducted the reaction pathways of the conversion of HMF to DMF over CoO(100)-O_V surface. The calculated E_a values of TS2 and TS1a can describe the importance of the active hydride species in the hydrogenolysis of HMF. The $H^{\delta+}$ attacks the O atom of $-CH_{2}O$ has the E_a of 1.42 eV, which is the rate-determining step of the HMF-to-BHMF reaction and the MHF-to-DMF reaction. The most difficult steps of the BHMF-to-HMMF and HMMF-to-DMF processes consume 1.3 eV and 1.23 eV, respectively. Can these mechanistic results be used to explain other results from experimental part (i.e. the results in Table 1)?
3. According to the description in the revised manuscript and Figure 13 in SI, the labels of the homolytic and heterolytic pathways on CoO(100)-O_V surfaces in Figure 5a seem to be wrong. Moreover, an arrow in Figure 5a should be removed.
4. In Figure 19 (in SI), the E_{ads} value should be changed to the relative energy (E_{rel}). The adsorption energy is calculated using the total energies of the isolated adsorbate(s) and the surface as references. However, there are different intermediates and gas species along the reaction pathways. The values given in Figure 19 are the relative energies compared with the initial state (at $E=0$ eV).
5. Are IM11 and FS configurations in Figure 19r and 19w wrong? They seem to be the TS4 structure in Figure 19m. Names of intermediates and TS states should be mentioned in the description on page 15-16 such as IM9 (the HMMF and water formation), and some TS states of important energy barriers. This would help readers to understand the pathway easily.
6. In Figure 16e and 16f, do those configurations have negative E_{ads} values (endothermic adsorption process)? Please carefully check those values and other typos in the manuscript and SI again.

Reviewer #3 (Remarks to the Author):

The authors have addressed my concerns regarding the manuscript and therefore i recommend to proceed with the publication of this interesting article.

Response to referee 2

Reviewer #2 (Remarks to the Author):

In the revised manuscript, the authors addressed all comments properly. However, there are some mistakes in the revised manuscript and SI. After reviewing this paper, I recommend that this paper is possible to be published in Nature Communications subjected to minor revision. Additional comments are listed below.

1. Scheme in Figure 1 in the revised manuscript is wrong. There is only hydrogenation (+H₂) in the HMF-to- BHMF step.

Reply: Thank you for your reminder, we have corrected them in the revised manuscript.

2. The authors conducted the reaction pathways of the conversion of HMF to DMF over CoO(100)-O_V surface. The calculated E_a values of TS2 and TS1a can describe the importance of the active hydride species in the hydrogenolysis of HMF. The H^{δ+} attacks the O atom of -CH₂~O has the E_a of 1.42 eV, which is the rate-determining step of the HMF-to-BHMF reaction and the HMF-to-DMF reaction. The most difficult steps of the BHMF-to-HMMF and HMMF-to-DMF processes consume 1.3 eV and 1.23 eV, respectively. Can these mechanistic results be used to explain other results from experimental part (i.e. the results in Table 1)?

Reply: We thank the reviewer for raising these concerns. The mechanistic results can really explain the experimental data in Table 1. Due to the thinner CoO films and the presence of metallic Co on the surface, Co₃O₄-300 and Co₃O₄-400 catalysts were difficult to provide sufficient and stable amounts of active H^{δ-} species and overcome the high energy barrier for the next step of hydrogenolysis reaction. As a result, Co₃O₄-300 and Co₃O₄-400 catalysts showed poor activity than that of Co₃O₄-250, with HMMF and BHMF being the main product, respectively. (see Page 16 in the revised manuscript)

3. According to the description in the revised manuscript and Figure 13 in SI, the labels of the homolytic and heterolytic pathways on CoO(100)-O_V surfaces in Figure 5a seem to be wrong. Moreover, an arrow in Figure 5a should be removed.

Reply: Thank you for your careful reminder, and we are sorry to make this mistake in

Figure 5a. We have corrected them in the revised manuscript.

4. In Figure 19 (in SI), the E_{ads} value should be changed to the relative energy (E_{rel}). The adsorption energy is calculated using the total energies of the isolated adsorbate(s) and the surface as references. However, there are different intermediates and gas species along the reaction pathways. The values given in Figure 19 are the relative energies compared with the initial state (at $E = 0$ eV).

Reply: We thank the reviewer for this suggestion. We have modified this section in the revised manuscript.

5. Are IM11 and FS configurations in Figure 19r and 19w wrong? They seem to be the TS4 structure in Figure 19m. Names of intermediates and TS states should be mentioned in the description on page 15-16 such as IM9 (the HMMF and water formation), and some TS states of important energy barriers. This would help readers to understand the pathway easily.

Reply: We thank the reviewer for this valuable suggestion, we feel very sorry for making this mistake and we have corrected them in the revised manuscript.

To make the article better understood by readers, we have modified Figure 6 and described the names of intermediates and TS states on page 15-16.

6. In Figure 16e and 16f, do those configurations have negative E_{ads} values (endothermic adsorption process)? Please carefully check those values and other typos in the manuscript and SI again.

Reply: Thank you for your careful reminder. We have checked these values, and these configurations (Supplementary Figure 16e and 16f) do have negative E_{ads} values.